# Volcanic Clouds Characterization of the 2020–2022 Sequence of Mt. Etna Lava Fountains Using MSG-SEVIRI and Products' Cross-Comparison

Lorenzo Guerrieri [1],* , Stefano Corradini [1] , Nicolas Theys [2] , Dario Stelitano [1] and Luca Merucci [1]

[1] Istituto Nazionale di Geofisica e Vulcanologia, Osservatorio Nazionale Terremoti, 00143 Rome, Italy; stefano.corradini@ingv.it (S.C.); dario.stelitano@ingv.it (D.S.); luca.merucci@ingv.it (L.M.)

[2] Royal Belgian Institute for Space Aeronomy (BIRA-IASB), 1180 Brussels, Belgium; nicolas.theys@aeronomie.be

\* Correspondence: lorenzo.guerrieri@ingv.it

**Abstract:** From December 2020 to February 2022, 66 lava fountains (LF) occurred at Etna volcano (Italy). Despite their short duration (an average of about two hours), they produced a strong impact on human life, environment, and air traffic. In this work, the measurements collected from the Spinning Enhanced Visible and InfraRed Imager (SEVIRI) instrument, on board Meteosat Second Generation (MSG) geostationary satellite, are processed every 15 min to characterize the volcanic clouds produced during the activities. In particular, a quantitative estimation of volcanic cloud top height (VCTH) and ash/ice/$SO_2$ masses' time series are obtained. VCTHs are computed by integrating three different retrieval approaches based on coldest pixel detection, plume tracking, and HYSPLIT models, while particles and gas retrievals are realized simultaneously by exploiting the Volcanic Plume Retrieval (VPR) real-time procedure. The discrimination between ashy and icy pixels is carried out by applying the Brightness Temperature Difference (BTD) method with thresholds obtained by making specific Radiative Transfer Model simulations. Results indicate a VCTH variation during the entire period between 4 and 13 km, while the $SO_2$, ash, and ice total masses reach maximum values of about 50, 100, and 300 Gg, respectively. The cumulative ash, ice, and $SO_2$ emitted from all the 2020–2022 LFs in the atmosphere are about 750, 2300, and 670 Gg, respectively. All the retrievals indicate that the overall activity can be grouped into 3 main periods in which it passes from high (December 2020 to March 2021), low (March to June 2021), and medium/high (June 2021 to February 2022). The different products have been validated by using TROPOspheric Monitoring Instrument (TROPOMI) polar satellite sensor, Volcano Observatory Notices for Aviation (VONA) bulletins, and by processing the SEVIRI data considering a different and more accurate retrieval approach. The products' cross-comparison shows a generally good agreement, except for the $SO_2$ total mass in case of high ash/ice content in the volcanic cloud.

**Keywords:** Mount Etna eruption; volcanic cloud top height; volcanic ash/ice particles and $SO_2$ retrievals

## 1. Introduction

By releasing large quantities of particles and gases into the atmosphere, volcanic eruptions can have a significant impact on human health [1,2], the environment [3–6], and climate [7–11] and pose a severe threat to aviation safety [12]. The residence time in the atmosphere of the emitted particles depends on their sizes and the height at which they are ejected. Typically, particles with a radius of less than about 20 μm can remain in the atmosphere for weeks and travel thousands of kilometers downwind. In addition to the particles, the most abundant gases are $H_2O$, $CO_2$, and $SO_2$ [13,14]. The water vapor in combination with the ash particles, which act as condensation nuclei, can form water particles that, under particular pressure and temperature conditions, turn into ice. Volcanic

ice clouds are spectrally indistinguishable from ice weather clouds but can be even more dangerous for aircraft as they can hide possible ash layers.

Etna (Sicily, Italy, 37.7°N; 15.0°E, Figure 1a), one of the most active volcanoes in the world, has exhibited since 2011 intense explosive activities ranging from Strombolian episodes to short lava fountain (LF) events [15–17]. In 2016, this eruptive regime gradually transitioned to mild Strombolian explosive activities of long duration associated with isolated episodes of lava flows from the volcano's summit craters [18]. The Christmas 2018 Etna eruption was preceded by a period of moderate explosive activity and small lava flows at the summit craters [19,20].

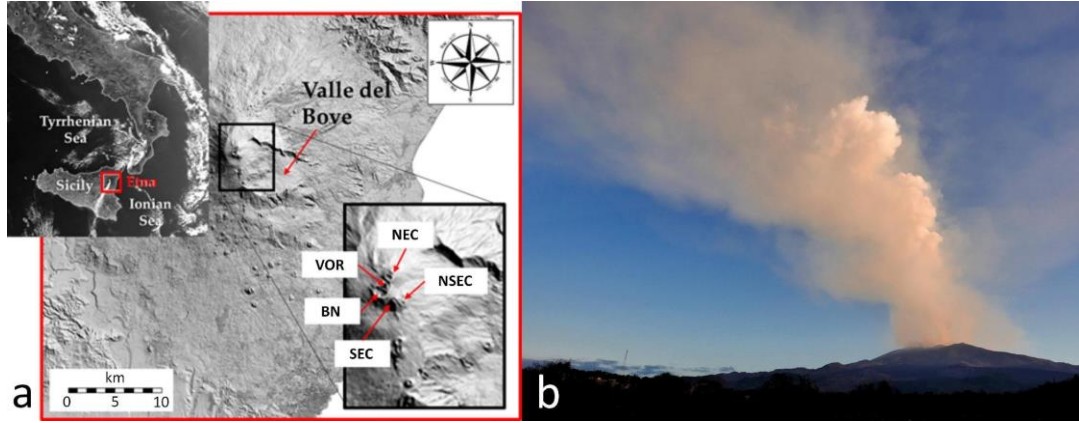

**Figure 1.** (**a**) Southern Italy image (from MODIS-Aqua data collected the 27 December 2018 at 12:20 UTC) and detail on Etna volcano summit craters (DEM from INGV Dataset "Tinitaly" https://tinitaly.pi.ingv.it/, accessed on 15 May 2019): Voragine (VOR), Bocca Nuova (BN), Northeast Crater (NEC), Southeast Crater (SEC), New SEC (NSEC) (figure modified from Corradini et al. [21,22]). (**b**) 25th Etna lava fountain activity in the early morning of 1 April 2021 (photo by L.G.).

On 13 December 2020, after about 18 months of eruptive pause, the volcano entered a new phase characterized by several LF episodes, all of them from the New Southeast Crater (NSEC) [23]. After three other events (21 and 22 December 2020 and 18 January 2021), from 16 February 2021, Etna produced a series of intense and frequent LFs (every ~1–2 days) characterized by several kilometers' high volcanic clouds and ash fallout in the surrounding areas of the volcano. This had a strong impact on viability, stability of the roofs, air traffic (the Catania airport is a major international hub), agriculture, water contamination, and on the health of the local population [23]. This first strong phase, with eight episodes in February and eleven episodes in March, finished with the 31 March–1 April 2021 LF (Figure 1b). After that, Mt. Etna took a break of about 50 days, and then a new event occurred on 19 May 2021. From this moment, the activity regained vigor with numerous fountains (eleven events in May, seventeen in June, seven in July) quite close in time but usually less energetic than the previous ones. Finally, the 2021 sequence came to an end with two episodes in August (9 and 29), one in September (21), and the last one on 23 October 2021. In 2022 only two LFs were observed on 10 and 21 February.

In this work, the data collected by the Spinning Enhanced Visible and InfraRed Imager (SEVIRI) instrument on board Meteosat Second Generation (MSG-4, Meteosat-11) geostationary satellite were used to fully characterize the volcanic clouds produced by all the different LFs by deriving their geometry (height and extension) and content (ash, ice, and $SO_2$). The complete list of LFs from December 2020 to February 2022 was derived from the Volcano Observatory Notices for Aviation (VONA) bulletin emanated from Istituto Nazionale di Geofisica e Vulcanologia-Osservatorio Etneo (INGV-OE) when the wording "lava fountain" was present. Overall, 59 lava fountains (see Table A1) were processed; regarding the 66 episodes reported in Calvari and Nunnari [24], we considered as a single LF the three close episodes of 13–14 December 2020 and the two close episodes of 22–23

February 2021, hardly distinguishable from SEVIRI. Furthermore, the four episodes of 27–28 May 2021 were excluded due to the strong presence of meteorological clouds in the area which did not allow volcanic cloud discrimination and detection.

The paper is organized as follows: in Section 2, the satellite, the atmospheric data, and the methods used for the volcanic height and particles/gas retrievals computation are described. In Section 3, all the results of the 2020–2022 Etna lava fountains sequence are reported: volcanic cloud heights, Brightness Temperature Differences (BTD) thresholds used for ash/ice particle discrimination, and total masses' time series of ash/ice/$SO_2$. In Section 3.4, the volcanic cloud heights and total masses are compared with the results obtained by other sensors, and a different retrieval procedure is applied to SEVIRI data. Finally, the conclusions are outlined in Section 4. In the Appendix are reported: a table summarizing all the LFs, and for each, the main results presented in the paper, the comparison with the LF heights and ground deposits, and a list of the acronyms used. As Supplementary Materials, the complete time series plots of VPR total masses of ash/ice/$SO_2$ obtained from the processing of more than two thousand SEVIRI images are displayed.

## 2. Materials and Methods

In this work, the data collected by SEVIRI were considered. SEVIRI has 12 spectral channels from visible (VIS) to Thermal InfraRed (TIR), a nadir spatial resolution of 3 km at sub-satellite point, and a temporal resolution of 5 or 15 min (Rapid Scan or Earth Full Disk, respectively). A satellite acquisition system (Multimission Acquisition SysTem, MAST) has been developed at INGV to collect and pre-process the SEVIRI images in real time [25]. All the SEVIRI images used in this work have been resized for a "Central Mediterranean Area" and resampled on a regular grid of $3 \times 3$ km$^2$. For each LF, the image processing starts when the eruption begins until the volcanic cloud is no longer detectable (by dilution or by leaving the considered area).

The atmospheric profiles, needed for the volcanic cloud altitude and ash/ice/$SO_2$ retrievals, were taken from the NCEP/NCAR Reanalysis Dataset of NOAA/ESRL Physical Sciences Laboratory (http://www.esrl.noaa.gov/psd/data/reanalysis/reanalysis.shtml, accessed on 12 April 2022 [26]). These data are distributed over global grids with $2.5° \times 2.5°$ spatial resolution, 17 pressure levels (1000, 925, 850, 700, 600, 500, 400, 300, 250, 200, 150, 100, 70, 50, 30, 20, 10 mbar) and surface level, 4-times daily (00, 06, 12, 18 UTC). These data are given in NetCDF format files of temperature, geopotential height, relative humidity, U and V wind components. From these global files, all the 2020–2022 vertical atmospheric profiles in the grid point closest to Etna (37.5°N, 15.0°E) were reconstructed.

### 2.1. Volcanic Cloud Top Height

The Volcanic Cloud Top Height (VCTH) is one of the most important parameters for aircraft security [25,27–29]. However, it also represents a key input for volcanic cloud retrieval procedures [30,31] and for a correct initialization of the volcanic ash dispersion and deposition models [32–35]. In this work, we compare the VCTH obtained from 3 different methods, all applied to SEVIRI images: darkest pixel, cloud tracking, and HYSPLIT.

#### 2.1.1. Darkest Pixel Method (DP)

The consolidated "Darkest Pixel" (DP) procedure is based on the comparison between the minimum SEVIRI 10.8 μm brightness temperature ($BT_{10.8}$) of the pixels contained in a fixed area over the summit craters and the atmospheric temperature profile measured in the same area and at the same time of satellite acquisition [30,36,37]. In this work, an area of $17 \times 17$ pixels around Etna (about $50 \times 50$ km$^2$) was chosen and the maximum value of the top altitude time series (obtained from several consecutive SEVIRI images) was taken as VCTH. The method is very simple and usually permits obtaining the VCTH with good accuracy. However, the DP method also has several weaknesses. The main one happens when the plume is not completely opaque. In this case, the radiance from the surface and the lower part of the atmosphere beneath the plume increases the top-of-atmosphere (TOA)

radiance measured by satellite, leading to underestimated plume heights. To consider the possible non-complete opacity of the pixel, the SEVIRI 10.8 μm brightness temperatures were decreased by 2 K as suggested by [36]. Another drawback of the method is due to the atmospheric temperature vertical behavior: the uncertainty on VCTH retrieval increases significantly around the tropopause because of the low temperature gradient and, due to the rise of stratospheric temperature, this often leads to double solutions (see Figure 2a). In this work, the lowest height has always been chosen, because injections of the plume into the stratosphere are quite unusual for the characteristics of the 2020–2022 Etna activity. The last drawback of the DP method is related to the possible thermal disequilibrium with the surrounding atmosphere. In this study, nine cases with a cloud top temperature lower than the minimum air temperature were found, resulting in no point of intersection with the atmospheric temperature profiles. For these specific cases, the nearest value (tropopause height) was considered as VCTH.

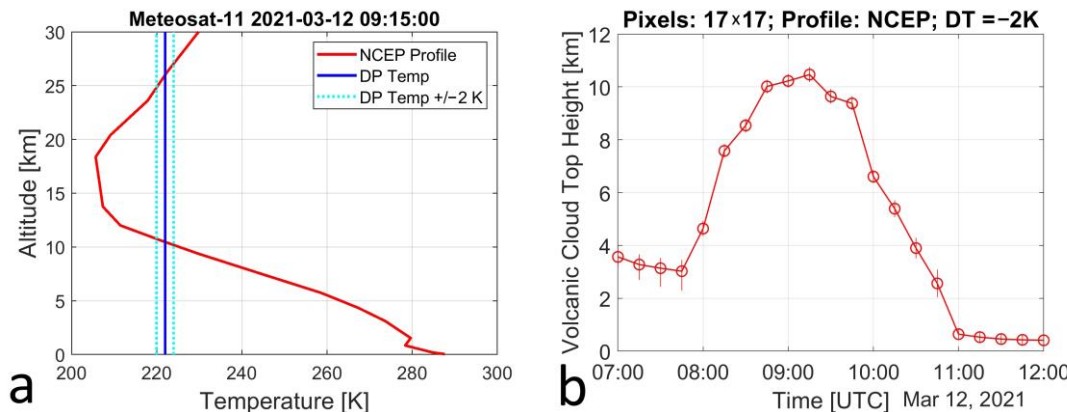

**Figure 2.** (**a**) Darkest Pixel (DP) procedure applied on the 12 March 2021, 09:25 UTC SEVIRI image. (**b**) The complete Volcanic Cloud Top Height (VCTH$_{DP}$) time series obtained for the 12 March 2021 Etna lava fountain.

The uncertainties of the DP method were estimated by considering $+/-2$ K of the value obtained from the coldest pixel temperature minus 2 K [36]. The mean uncertainty for all the LFs considered is $+/-0.3$ km.

Figure 2 shows an example of the VCTH estimation from the DP procedure applied on the 12 March 2021 LF. In the left plot, the volcanic cloud's coldest pixel temperature value with its uncertainties (blue vertical solid and dashed lines, respectively), obtained from the SEVIRI image collected at 09:25 UTC, is compared with the NCEP temperature profile collected in the same region at the closest time (12 March 2021 12 UTC at 37.5°N, 15.0°E, red solid line). In the right plot, the VCTH time series obtained by processing the different SEVIRI images of the same day every 15 min is displayed. The NCEP temperature profiles of 12 March 2021 at 06 and 12 UTC were used for SEVIRI images before and after 9 UTC, respectively.

### 2.1.2. Cloud Tracking Method (CT)

The high data frequency of SEVIRI images can be exploited to retrieve wind speed and direction of the volcanic clouds for each event [37,38]. First of all, it is necessary to perform the detection of the volcanic cloud (see Section 2.2). In case of an intense but short-lived eruption (such as LFs), it is quite easy to find and track, in the series of images, the pixels with the minimum radiance at 10.8 μm, which corresponds to the maximum ash/ice concentration. Computing the distance from the top of the volcano for at least 2–3 h from the start of the eruption and using a linear fit (blue line), the speed of the volcanic cloud is obtained (see Figure 3a). Finally, by comparing the retrieved peak speed with the wind speed of an atmospheric profile collected at the same time and position (red line, Figure 3b), the volcanic cloud altitude can be derived. It is important to note that, unlike the

DP method, this procedure does not necessarily produce the maximum height (VCTH), but depending on the location of the center of mass, a lower altitude may result. Unfortunately, due to the characteristics of atmospheric profiles, much more than a single intersection point can often be found. When this happens, the use of wind direction can provide help, even though sometimes only a range of height can be obtained; in these cases, the mean value was considered. Other sources of uncertainty are the non-linearity of the plume trajectory due to the non-uniformity of the wind field, which can produce large errors with this method.

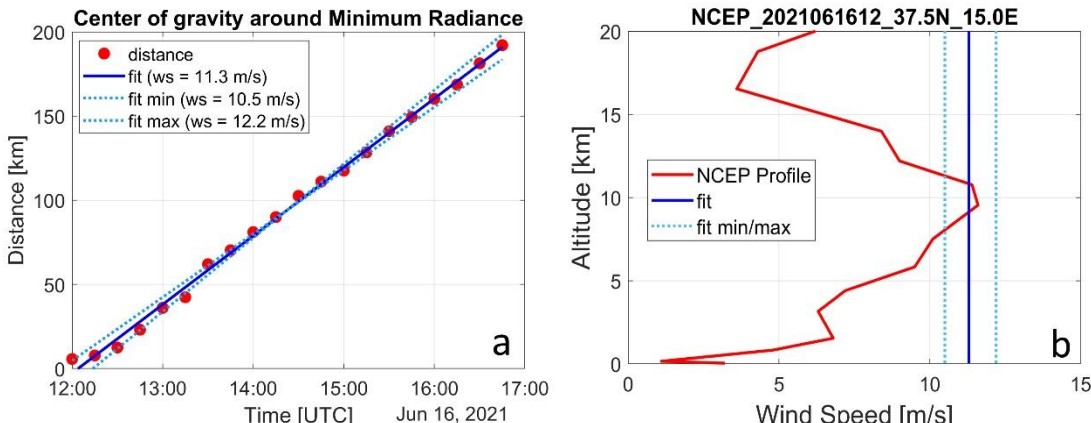

**Figure 3.** Example of VCTH computation using the Cloud Tracking (CT) procedure applied on the 16 June 2021 Etna lava fountain: (**a**) computation of mean wind speed (**b**) comparison with an atmospheric profile collected in the same region at the closest time.

The uncertainties of the CT method were computed considering the standard error of the estimate of the linear fit (cyan lines of Figure 3). The mean uncertainty for all the LFs considered is +/−0.7 km.

### 2.1.3. HYSPLIT Method (HY)

Finally, a similar procedure based on the Hybrid Single-Particle Lagrangian Integrated Trajectory (HYSPLIT) model has been also considered [39–41]. By plotting several forward trajectories (obtained from the GFS-0.25° archive) at different altitudes starting from the top of Etna at the time of the eruption start, it is possible to visualize which one better corresponds to the volcanic cloud position for a SEVIRI image collected some hours later. In its basic form, the HYSPLIT model does not consider the particulate transport, so formally it would only be suitable for deriving the height of the $SO_2$ cloud. Regardless, the HYSPLIT trajectories tool is a model widely used for the estimation of the volcanic cloud altitudes under the main hypothesis that ash, ice and $SO_2$ are totally collocated [37,42,43]. In this case, too, the characteristics of wind speed and direction can produce large uncertainties if there is not a marked vertical wind shear. On the other hand, the dispersion of the volcanic cloud can often highlight the different heights of the various parts of the plume. In both these cases it was assumed the maximum height was VCTH. The forward trajectories were chosen at 1 km altitude steps, so the uncertainty of the HY method was assumed +/−0.5 km.

Figure 4 shows an example of the HY method applied on the 9 August 2021 Etna LF. The 12 h forward trajectories start from 02:00 UTC (beginning of the eruption) at 8 km (red), 9 km (blue), and 10 km (green) above sea level. The corresponding SEVIRI image (at 14:00 UTC) highlights the presence of the volcanic cloud (contoured in yellow) in the Mediterranean Sea, between 35°N–36°N and 18°E–20°E. From the comparison between the trajectories and the volcanic cloud position, the VCTH retrieved is about 9 km.

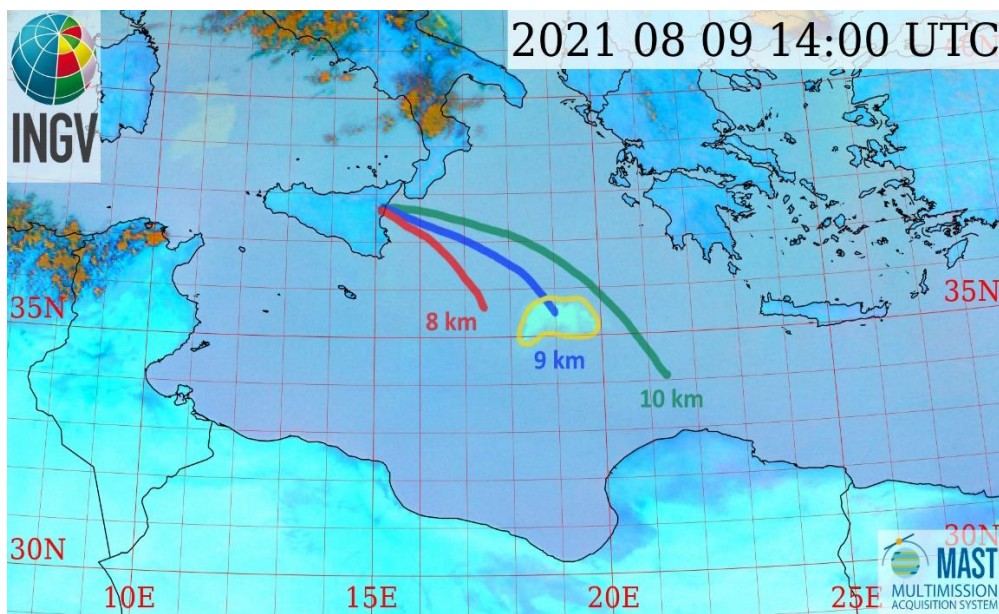

**Figure 4.** Example of VCTH computation using the HY procedure applied on the 9 August 2021 Etna lava fountain. SEVIRI image collected at 14:00 UTC, RGB composition (R: $BT_{12.0}-BT_{10.8}$; G: $BT_{10.8}-BT_{8.7}$; B: $BT_{10.8}$) with the volcanic cloud outlined in yellow. The output of the HYSPLIT model considering three forward trajectories at different altitudes (8, 9, and 10 km asl) is overlaid on the figure.

### 2.2. Volcanic Cloud Detection and Discrimination

In this work, the volcanic cloud detection was performed by visual inspection of an RGB composition obtained using a combination of the BTs of the channels centered at 8.7, 10.8, and 12 μm ($BT_{8.7}$, $BT_{10.8}$, $BT_{12}$) that allows identifying both the volcanic cloud particles and $SO_2$. For each SEVIRI image, a "plume mask" was obtained drawing a region of interest (ROI) around the volcanic cloud. Once the cloud has been identified, it is also necessary to discriminate the types of particles from which it is mainly composed. For this task, the BTD approach was used [44,45]. This method allows discriminating between ash and ice/water particles by exploiting the different spectral absorption in the TIR spectral range. In this interval, the absorption of ash particles at 10.8 μm is larger than that at 12 μm. The opposite happens for ice/water clouds, which absorb more significantly at longer TIR wavelengths. Therefore, the BTD, defined as the difference between the brightness temperature computed at 10.8 and 12 μm ($BT_{10.8}-BT12$), turns out to be generally negative (BTD < 0) for the region affected by ash and positive (BTD > 0) for the region containing ice/water clouds. In almost all the LFs considered, the formation of high quantities of water/ice particles (strongly positive BTD values) has been observed. Due to the high altitudes generally reached by the LF volcanic clouds, we considered the formation of ice much more likely than the presence of water droplets in liquid form.

Figure 5 shows an example of RGB composition and BTD. In the RGB image (panel a), the presence of volcanic ash, ice, and $SO_2$ is identified by the red, dark, and light green colors, respectively, while the b and w BTD image (panel b) indicates the existence of both ash (dark part, negative values) and ice (white part, positive values) particles.

The BTD approach is effective and simple to apply, even if it can lead to false positive ash detections (pixels wrongly detected as ashy) in some cases such as on clear surfaces during the night, on deserts, on very cold or ice surfaces, etc. [46]. Otherwise, the BTD can lead to false negative ash detection (pixels wrongly detected as non-ashy) that may arise in case of high-water vapor content. This condition, frequent on Etna volcano, can hide and then cancel out the ash particles' effects on the BTD, thus revealing fewer ashy pixels than those that exist [46,47]. For this reason, a correction, based on Radiative Transfer Models (RTM) computations, has to be applied [36,47]. Figure 6a shows an example of the inverted

arches curves obtained from ice (cyan lines) and ash pumice type (black lines, [48]) using MODTRAN 5.3 [49]. In this plot, the Aerosol Optical Depth at 550 nm ($AOD_{0.55}$) varies from 0 (right, bigger $BT_{10.8}$ values) to 10 (left, lower $BT_{10.8}$ values) for both ash and ice, while Effective Radius ($R_e$) varies from 0.55 to 10 µm (down to up) for ash and from 1.39 to 50 µm (up to down) for ice. To obtain a BTD threshold to use for discriminating between ash and ice, a quite good approximation is to consider a horizontal linear fit (red line) of the points with maximum $R_e$ (10 µm for ash and 50 µm for ice). Figure 6b shows the trend of the BTD thresholds (obtained as just described) as a function of the months of the year, VCTH (4.5–12.5 km), and the View Zenith Angle (VZA) whose typical values for SEVIRI "Central Mediterranean Area" are 35°–55° plotted as error bars. Specifically, 12 monthly average (years 1981–2010) atmospheric profiles (PTH) and sea surface temperatures (SST) computed from NCEP/NCAR database for the South Italy area (32.5°–42.5°N; 10.0°–20.0°E) have been considered and used in MODTRAN 5.3 simulations. The BTD dependence on VZA (not shown in figure) is quite linear and directly proportional: higher VZA produce higher BTD thresholds and vice versa.

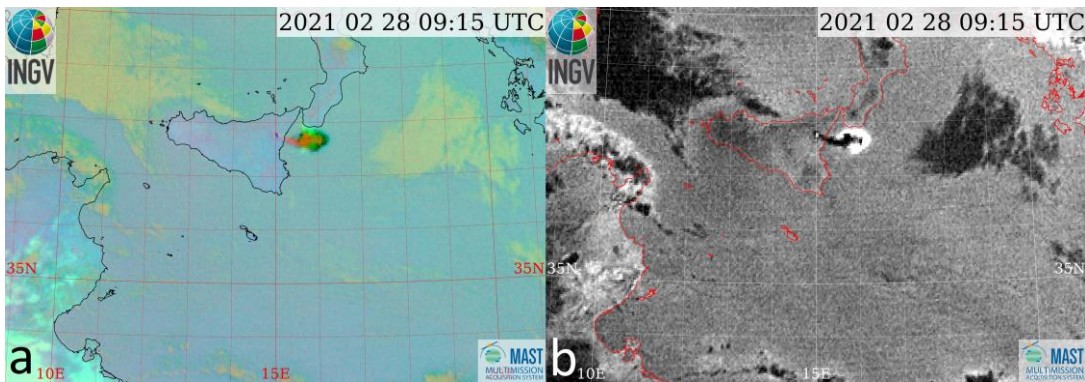

**Figure 5.** SEVIRI image collected at 09:15 UTC, 28 February 2021. (**a**) RGB composition (R: $BT_{12.0}-BT_{10.8}$; G: $BT_{10.8}-BT_{8.7}$; B: $BT_{10.8}$). (**b**) Brightness Temperature Differences (BTD = $BT_{10.8}$-$BT_{12}$).

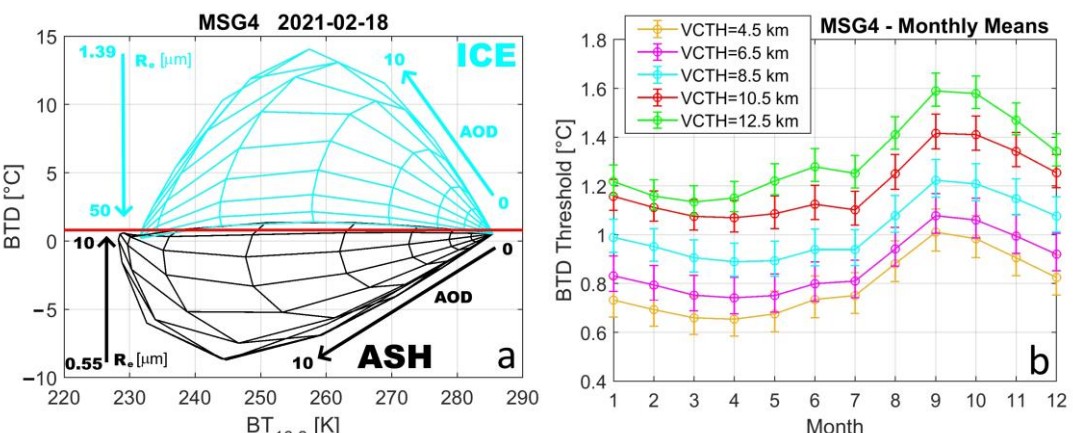

**Figure 6.** (**a**) Inverted arches curves of ice (cyan lines) and ash (black lines) obtained for MSG4-SEVIRI 18 February 2021 Etna LF using MODTRAN 5.3 RTM and varying $AOD_{0.55}$ and $R_e$. The BTD threshold (BTD = 0.8) is obtained with a horizontal linear fit (red line) of the points with maximum $R_e$. (**b**) Theoretical MSG4-SEVIRI BTD thresholds monthly variation as a function of VCTH of Etna and the VZA (identified by error bars).

As Figure 6b clearly shows, the combination of the various parameters (PTH, SST, VCTH, and VZA) can contribute significantly to the variation of the BTD threshold (from 0.6 to 1.7). For this reason, the BTD values used in this work were obtained from RTM simulations specific for SEVIRI (MSG4) for each 2020–2022 LF episode (see Section 3.2).

### 2.3. Particles and SO₂ Retrieval Method

The Volcanic Plume Retrieval (VPR) procedure is a linearization of the radiative transfer equation capable to retrieve from multispectral satellite images the $AOD_{0.55}$, the $R_e$, the columnar abundance (m), and the total mass (M) of particles (ash, ice, water droplets, etc.) contained in a volcanic cloud [31,38,50]. The channels used for the volcanic particles' retrievals are those centered at 10.8 μm and 12 μm, while for the $SO_2$ estimation, the channel centered at 8.7 μm is considered. Because of the ash/ice particles' absorption in the whole TIR spectral range, their quantitative estimations are taken into account to correct for the $SO_2$ amounts. The lack of this correction would otherwise lead to a significant overestimation of the amount of $SO_2$ present in the volcanic cloud [51].

The only VPR input required at run time is the plume temperature which can be easily obtained from a relevant atmospheric profile knowing the plume altitude. Before running, VPR also needs a "plume mask" so volcanic cloud detection and discrimination (ash or ice) is necessary. The main advantage of the VPR procedure is that it is easy to use and very fast. Coupled with SEVIRI's high temporal resolution, it allows for quick and reliable volcanic cloud retrievals during both day and night.

Several comparisons have been made in the past between the VPR and other procedures based on the calculation of atmospheric corrections, showing good agreement [22,52]. Among the particle's properties, $R_e$ is that with the lower accuracy, while a better performance is related to $AOD_{0.55}$ and M. When no particles are present in the volcanic cloud or if the aerosol transmittance would be perfectly known, the $SO_2$ accuracy is very high; unfortunately, this excellent result is reduced in most real cases since the $SO_2$ retrieval is highly dependent on that of the ash/ice particles. In particular, the presence of high quantities of ice can produce greater $SO_2$ uncertainties [50]. In this work, we considered an overall error of ±40% for particles (ash/ice) and ±50% for $SO_2$ total mass, due to the presence of large amounts of ice.

### 2.4. Other Methods Used for Cross-Comparison

#### 2.4.1. VONA

The column height reported in the INGV VONA bulletin (www.ct.ingv.it/index.php/monitoraggio-e-sorveglianza/prodotti-del-monitoraggio/comunicati-vona, accessed on 16 March 2022) was considered [37,53]. This value is predominantly obtained from visible surveillance cameras placed around Etna (during daytime and with good visibility conditions) or, more occasionally, from satellites (darkest pixel procedure). Unfortunately, in some cases, the VCTH is not given, while when more than one value was present (due to LF progress), the maximum was considered. The uncertainty of the VONA column height was set to ±0.5 km, according to Scollo et al. [53].

#### 2.4.2. TROPOMI

The TROPOspheric Monitoring Instrument (TROPOMI) is a spectrometer on board Sentinel-5 Precursor (S5P) polar orbit satellite that covers a spectral range from ultraviolet (UV) to short wave infrared (SWIR), with a spatial resolution of $5.5 \times 3.5$ km$^2$ and a revisit time of about 1 day [54]. The UV channels were used for the validation of volcanic cloud altitude and $SO_2$. Particularly significant for TROPOMI is the ability to detect $SO_2$ columnar abundance of about 0.02 g/m$^2$ (0.7 DU), i.e., about thirty times better than the sensitivity of the multispectral satellite sensors [22]. The plume height of $SO_2$ was retrieved from TROPOMI measurements using the algorithm of [55]. In brief, the radiance measurements are analyzed using a Covariance-Based Retrieval Algorithm (COBRA) in which the $SO_2$ vertical column and plume height are jointly retrieved based on a look-up table approach. The $SO_2$ total mass is obtained by summing all contributions from the pixels of the plume. Note that by doing so, some pixels are rejected (those for which the height retrieval did not converge, essentially for vertical columns < 5 DU), and therefore the $SO_2$ total mass is probably a lower estimate. The uncertainty of the TROPOMI height and $SO_2$ total mass

was set to $+/-2$ km [55] and 35% [22], respectively, an error estimate which might be too optimistic for conditions with a lot of volcanic ash.

### 2.4.3. $LUT_p$

The Look-Up Tables procedure ($LUT_p$) is a well-known particles and gases retrieval method [30,45,47,51,56]. It uses the same three TIR SEVIRI channels as VPR (8.7, 10.8, and 12 μm) and, through linear and bilinear interpolation with specific RTM simulations, permits to obtain particle $AOD_{0.55}$, $R_e$, mass, and $SO_2$ mass of a volcanic cloud. Generally, the $LUT_p$ is considered more accurate than the VPR [52], but this comes at the cost of a greater number of input parameters and a much longer calculation. For validating the VPR results, the $LUT_p$ was applied to the same SEVIRI images using the same RTM simulations used for the BTD thresholds estimation (see Section 3.2) plus 11 $SO_2$ values (from 0 to 10 g/m$^2$, step 1 g/m$^2$). An overall error of $\pm40\%$ for particles and $SO_2$ total mass was considered [21,30].

## 3. Results

### 3.1. Volcanic Cloud Top Height

Figure 7 shows the VCTH for all the 2020–2022 Etna LFs using the three different procedures described in Section 2.1 (DP, CT, and HY). Each method has strengths and weaknesses depending on the conditions, so to get the "best estimate", a weighted average is presented (see Equations (1)–(3)). The choice of the weights derives from the intrinsic characteristics of the three methods and from our experience: DP is considered the most reliable for high altitudes (strong eruptions with completely opaque pixels), while HY turns out better for low altitudes (weak eruptions). Because the CT method and the HYSPLIT model use a single atmospheric profile and a complete set of meteorological data, respectively, the latter is considered generally more reliable. The overall VCTH uncertainty was obtained from the combination of the uncertainties (according to Equations (1)–(3)) of the three methods used, resulting in a mean value of $+/-0.5$ km.

$$VCTH = 0.5 \cdot VCTH_{DP} + 0.2 \cdot VCTH_{CT} + 0.3 \cdot VCTH_{HY} \text{ for } VCTH_{DP} > 9 \text{ km} \tag{1}$$

$$VCTH = 0.4 \cdot VCTH_{DP} + 0.2 \cdot VCTH_{CT} + 0.4 \cdot VCTH_{HY} \text{ for } 6 \leq VCTH_{DP} \leq 9 \text{ km} \tag{2}$$

$$VCTH = 0.2 \cdot VCTH_{DP} + 0.2 \cdot VCTH_{CT} + 0.6 \cdot VCTH_{HY} \text{ for } VCTH_{DP} < 6 \text{ km} \tag{3}$$

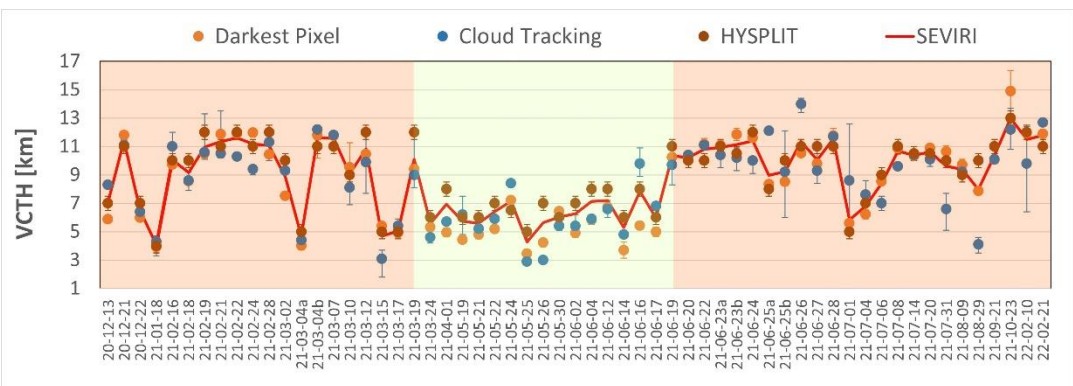

**Figure 7.** Time series of the VCTH obtained using the DP, CT, and HY procedures for all the 2020–2022 Etna LFs. The error bars represent the uncertainties of the three methods as described in Section 2.1. The red solid line indicates the VCTH weighted mean (Equations (1)–(3)) used as input in the VPR procedure. The "Date" labels (*x*-axis) are in the format year-month-day, with letters a and b to distinguish the LFs that occurred on the same day.

Generally, the three methods agree well with each other: the mean absolute difference is 1.1, 1.2, and 1.4 km for DP vs. CT, DP vs. HY, and CT vs. HY, respectively. The VCTH ranges from 4 km (18 January 2021) to 13 km (23 October 2021). As the figure shows, the results emphasize three main periods (highlighted by the background colors of the plot): after an initial phase with low altitudes volcanic cloud (except for 21 December 2020), the period from 16 February to 19 March 2021 was characterized by high volcanic plumes with 10 episodes with VCTH greater than 10 km. Then, a long phase (3 months) occurred with 15 episodes with lower VCTH that ranges between 5–7 km, and from 19 June 2021, a new VCTH increase with several values greater than 10 km. Finally, this long eruptive phase ended with four strong (but spaced over a time range of 5 months) paroxysms from 21 September 2021 to 21 February 2022 with VCTH greater than 11 km. The complete list of VCTH values is given in Table A1.

It is also interesting to note that the retrieved SEVIRI-VCTH trend is in good agreement with the LF maximum heights trend shown by Calvari and Nunnari [24], obtained by exploiting the measurements collected from the ground-based TIR cameras (see Figure A1), with a Pearson's correlation coefficient of 0.65. The average difference between VCTH and LF maximum heights is 3.5 km (considering that NSEC is about 3.3 km asl high).

### 3.2. BTD Thresholds

The BTD threshold values used in this work to discriminate ashy and icy pixels were obtained from specific RTM simulations related to each LF considered as described in Section 2.2. The 2020–2022 atmospheric vertical profiles (PTH) at (37.5°N, 15.0°E) from NCEP/NCAR dataset were used; depending on eruption timing (nighttime or daytime), the 00 UTC or 12 UTC was chosen. From each of them, the total precipitable water (PW) was computed, too. SST values were obtained from "NOAA Daily Optimum Interpolation Sea Surface Temperature" (https://www.psl.noaa.gov/data/gridded/data.noaa.oisst.v2.highres.html, accessed on 19 July 2022 [57–59]). These data are provided at a 0.25° × 0.25° spatial resolution and were averaged over an area between 35°–40°N and 12.5°–17.5°E. Sea surface emissivity was fixed to 0.98. The VCTH weighted mean values (see Section 3.1, Table A1) were considered, with a constant plume thickness of 1 km. Pumice ash [48] and ice optical properties (extinction coefficient, single scattering albedo, and asymmetry parameter) derive directly from the application of the Mie theory considering a specific particle's refractive index and size distribution. The refractive indexes derive from the Aerosol Refractive Index Archive-ARIA database, compiled from the EODG group of Oxford University (http://eodg.atm.ox.ac.uk/ARIA, accessed on 27 October 2022), while the size distribution is considered log-normal. Regarding the VZA, a fixed value was considered for a geostationary SEVIRI sensor (VZA = 45°, typical of the Etna area). In this way, for each LF considered, only one BTD threshold value was used.

In Figure 8c, all the SEVIRI-BTD values used are reported. They range between 0.48 and 1.83 with a mean (standard deviation) value of 1.04 (0.33). The major contribution to BTD variations is due to VCTH (Figure 8b), even if PW and SST (see Figure 8a) are significant, too. Sometimes similar VCTH values do not correspond to the same BTD values. For example, the 21 December 2020 and 10 February 2022 LFs have quite the same VCTH (11.4 and 11.5 km, respectively) and very different BTD (1.7 and 1.0, respectively). For SEVIRI, the Pearson's correlation coefficient between BTD thresholds and VCTH, PW, and SST is 0.73, 0.57, and 0.27, respectively. Given the wide temporal extension (more than 1 year) and the large range of volcanic cloud heights considered (from 4 to 13 km), the average value BTD = 1 can be considered a first good approximation for Etna, instead of the standard BTD = 0.

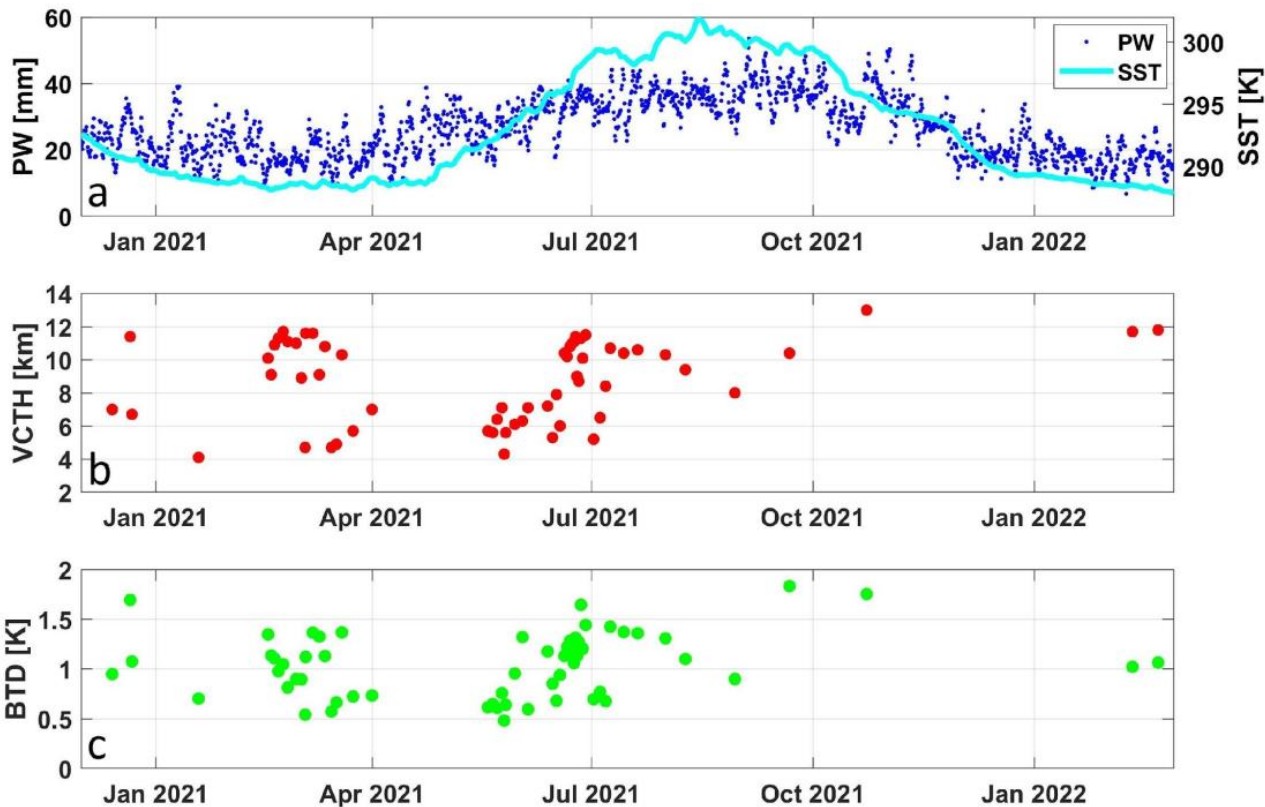

**Figure 8.** 2020–2022 Etna LF time series of: (**a**) Precipitable Water (PW, [mm]) and Sea Surface Temperature (SST, [K]). (**b**) Volcanic Cloud Top Height (VCTH, [km]). (**c**) BTD threshold used to distinguish between ashy and icy pixels.

*3.3. Total Mass of Ash, Ice, and SO$_2$*

Once the VCTH has been computed and the volcanic cloud detection and discrimination performed, the VPR procedure can be applied. From each satellite image, maps of ash/ice R$_e$ [μm], AOD$_{0.55}$ and m$_a$/m$_i$/m$_{SO2}$ [g/m$^2$] were computed. Finally, M$_a$/M$_i$/M$_{SO2}$ [Gg] total masses were easily calculated by adding the contribution of all the pixels multiplied by their area.

Figure 9 shows an example of columnar abundance maps and time series of M$_a$/M$_i$/M$_{SO2}$ for 23 October 2021 LF. The time trend highlights an almost simultaneous growth of both ash and ice from about 9 UTC until 10 UTC. After that, a decrease in ash and an increase in ice quantities are observed, compatible with the progressive ice generation due to the ash particles that act as condensation nuclei. The SO$_2$ presents a strong increase until about 10:15 UTC (when the LF ends, represented by the dashed red vertical line) and then a slight increase until 13:30 UTC. This latter effect is probably not a real increase in SO$_2$ but a consequence of the huge growth of the ice content that causes some criticalities in the retrieval correction procedure. The following decrease in SO$_2$ mass is due to the volcanic cloud dilution and chemical recombination. It should be noted that the dilution of the ice/ash particles (decrease in the total mass) usually begins earlier and is faster (the descent is steeper) than that of SO$_2$, which remains in the atmosphere for a longer time. The complete VPR total masses' time series for all the LFs considered can be found in the Supplementary Materials (Figure S1).

In Figure 10, the VPR M$_a$/M$_i$/M$_{SO2}$ for all the LFs were reported (in logarithmic scale). For each LF, the maximum value obtained from the time series was considered. To avoid the anomalous SO$_2$ increase after the end of LF, the SO$_2$ maximum value in the time range between the start and the end (+1 h) of the eruption was considered, in which the LF end time was obtained from Calvari and Nunnari [24]. In most cases, the ice content

is greater than ash but there are still some episodes where the opposite occurs, both with low and high plume heights. Together with $M_a/M_i/M_{SO2}$, Figure 10 also reports the VCTH to emphasize the general good correlation between the height and total masses as documented in Mastin et al. [60] where different studies on this topic are analyzed. The Pearson's correlation coefficient between VCTH and the logarithm of $M_x$ is 0.75 (ash), 0.64 (ice), and 0.57 ($SO_2$). Finally, note how the generation of ice is related not only to the volcanic cloud height but also to the season: during the summer (yellow background color), with almost the same plume height, the amount of ice is lower than that observed in winter (green background color).

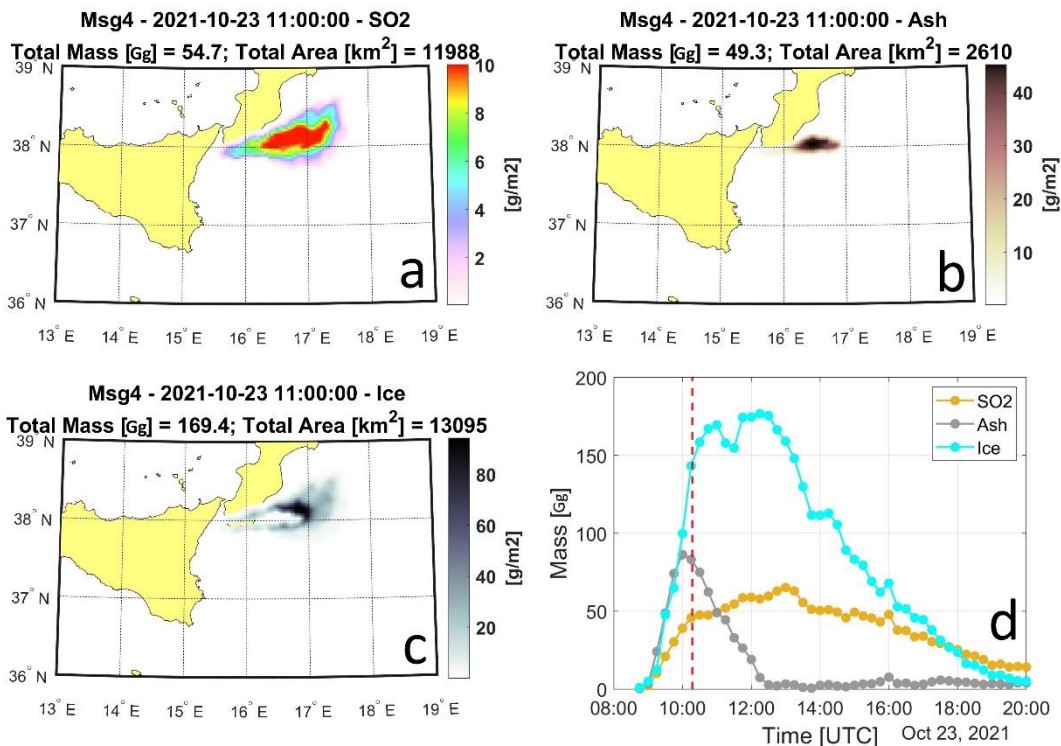

**Figure 9.** VPR outputs for 23 October 2021 Etna lava fountain: (**a**) $SO_2$, (**b**) ash, and (**c**) ice columnar abundance [g/m$^2$] retrieval maps. (**d**) Ash/ice/$SO_2$ total mass [Gg] time series. The red dashed vertical line represents the end of the LF, obtained from Calvari and Nunnari [24].

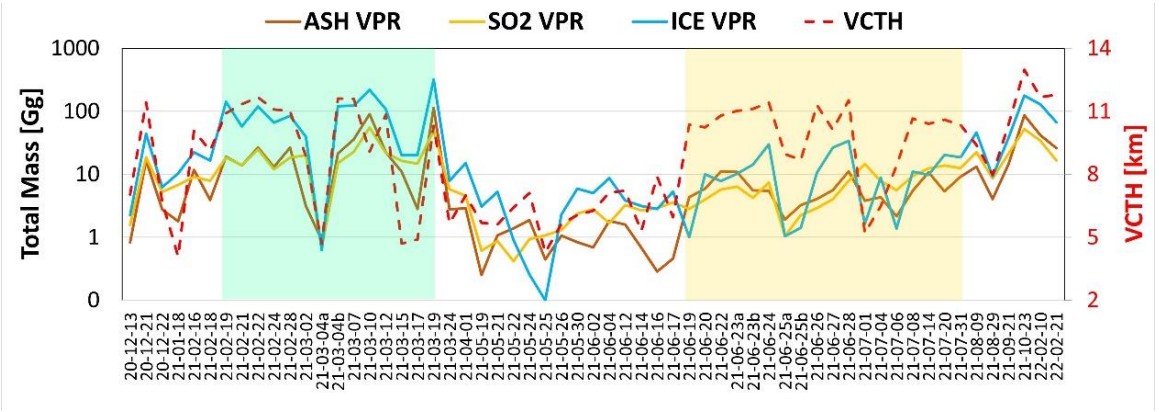

**Figure 10.** VPR ash/ice/$SO_2$ total masses and VCTH time series for all the 2020–2022 Etna LFs. The "Date" labels (*x*-axis) are in the format year-month-day, with letters a and b to distinguish the LFs that occurred on the same day.

The VPR total mass results have also been compared with the lava deposits on the ground obtained by Ganci et al. [61] exploiting the same SEVIRI images (Figure A2). The two different products show quite good agreement, with a Pearson's correlation coefficient between the lava volume and the logarithm of ($M_a$ + $M_i$ + $M_{SO2}$) of 0.57.

*3.4. Products Cross-Comparison*

Figure 11 shows the cross-comparison between SEVIRI, INGV-VONA bulletin, and TROPOMI VCTHs. Generally, TROPOMI heights are lower than those obtained from SEVIRI and surveillance cameras. However, there is a fundamental difference: the VCTH-DP and VONA estimates are related to aerosols emissions near the volcano (at the beginning of the eruption), while TROPOMI height refers to $SO_2$ which corresponds to a rather well-dispersed plume at quite some distance from Etna. Dilution of the cloud is certainly important in this case. Moreover, the effect of ash on the measurement sensitivity of TROPOMI can lead to biases in the retrieved $SO_2$ heights. Overall and considering all these different characteristics, the agreement of the various trends is quite good. Quantifying the differences between SEVIRI-VCTH and TROPOMI height, we find large discrepancies (>3 km) only for a few LFs (8 out of 40): 16 February, 24 and 27 June, 6 and 14 July, 29 August, 23 October in 2021, and 21 February 2022. The dilution effect (S5P passage is more than 12 h after the LF climax time) is probably the explanation for the 16 February, 6 July, and 29 August 2021 cases, while the high presence of ash/ice could be the cause of the TROPOMI underestimation for the other cases.

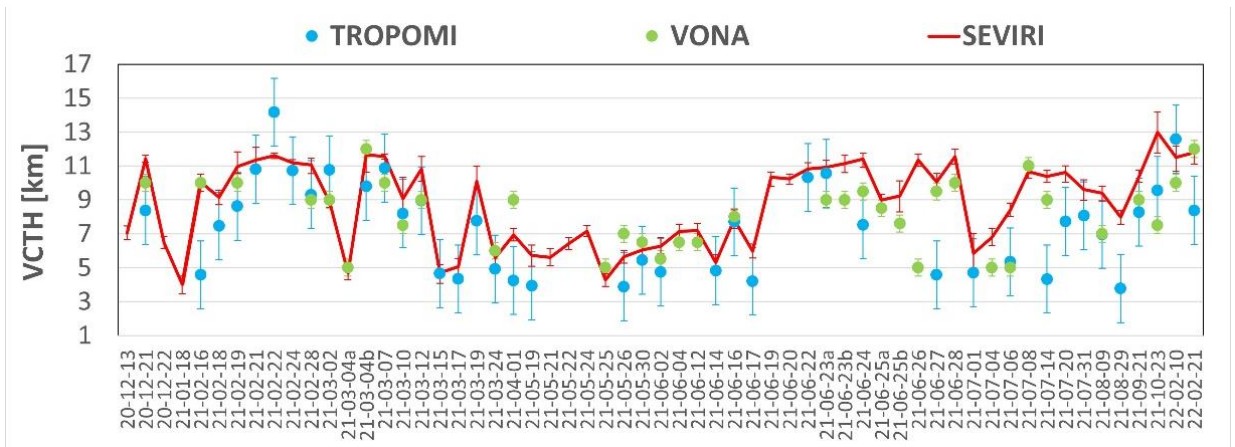

**Figure 11.** Cross-comparison between the VCTH obtained from SEVIRI as the weighted mean of the DP, CT, and HY procedures, TROPOMI $SO_2$ altitude, and VONA reports. The error bars represent the height uncertainties of the three sensors. The "Date" labels (*x*-axis) are in the format year-month-day, with letters a and b to distinguish the LFs that occurred on the same day.

Figure 12 shows the scatter plots between VPR and $LUT_p$ total masses for all the 59 LFs considered. The two procedures appear in good agreement for ash and ice retrievals, while for $SO_2$ the differences are significant in some cases. $SO_2$ $LUT_p$ results are generally lower than VPR, especially for the bigger activities, when high quantities of ash/ice are present in the volcanic cloud. In these cases, VPR $SO_2$ retrievals are characterized by bigger uncertainties, as described in Section 2.3, thus probably give out an overestimated value for the $SO_2$ total mass. However, all the VPR and $LUT_p$ ash/ice values agree (within the errors), while regarding $SO_2$ only one (10 March 2021) is outside the limits. $R^2$ values are 0.94, 0.93, and 0.14 for ash/ice/$SO_2$, respectively, but this latter becomes 0.79 just neglecting the nine LFs with ice total mass greater than 100 Gg. The complete list of VPR and $LUT_p$ total masses is given in Table A1.

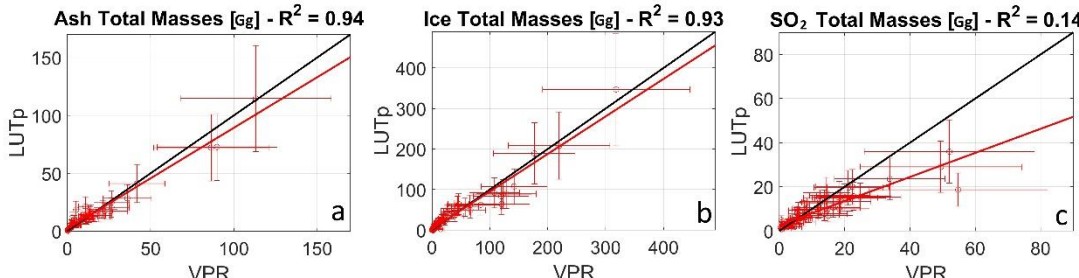

**Figure 12.** Comparison between SEVIRI (**a**) ash, (**b**) ice, and (**c**) SO$_2$ total masses computed using VPR and LUT$_p$ approaches.

In Figure 13, SO$_2$ total masses from SEVIRI and TROPOMI are shown. The temporal correspondence between the various images was obtained considering a maximum time difference of $+/-1$ h (but usually is less than 10 min), resulting in only 22 LFs. On the secondary axis (right), the sum of ash and ice total mass is reported also to highlight the presence or not of large quantities of particles inside the volcanic cloud. Generally, the agreement is quite good for weak eruptions while it is poor for bigger eruptions where SEVIRI SO$_2$ results are found to be greater than TROPOMI. These paroxysms were characterized by a high particle (ash/ice) content (dashed line) that certainly can affect all the measurements.

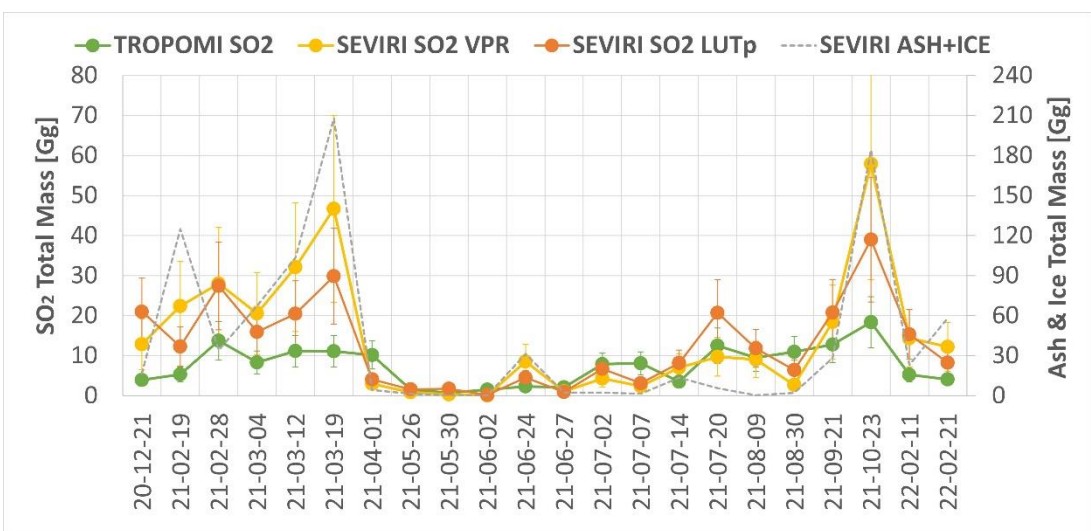

**Figure 13.** SO$_2$ total masses from SEVIRI-VPR, SEVIRI-LUT$_p$, and TROPOMI. The error bars represent the retrieval uncertainties of the three procedures. On the secondary axis (right), the sum of VPR ash and ice total mass is reported also as a reference. The "Date" labels (*x*-axis) are in the format year-month-day.

Figure 14 shows the comparison between SO$_2$ column densities derived from SEVIRI (both with VPR and LUT$_p$ procedures) and from TROPOMI measurements. Two LFs were considered: the strong eruption of 23 October 2021 in which SEVIRI SO$_2$ total mass is much greater that from TROPOMI (first row) and the weak one of 9 August 2021 in which the two sensors almost agree (second row). In these two examples, the spatial extent of the volcanic clouds is almost the same, while in the case of 23 October 2021, the SO$_2$ column densities are significantly higher for SEVIRI than for TROPOMI, especially in the central part of the volcanic cloud. The reason for that is the presence of a high concentration of ash and ice particles that cause an overestimation and an underestimation of the columnar SO$_2$ content when retrieved in the TIR and UV spectral ranges, respectively [21,22].

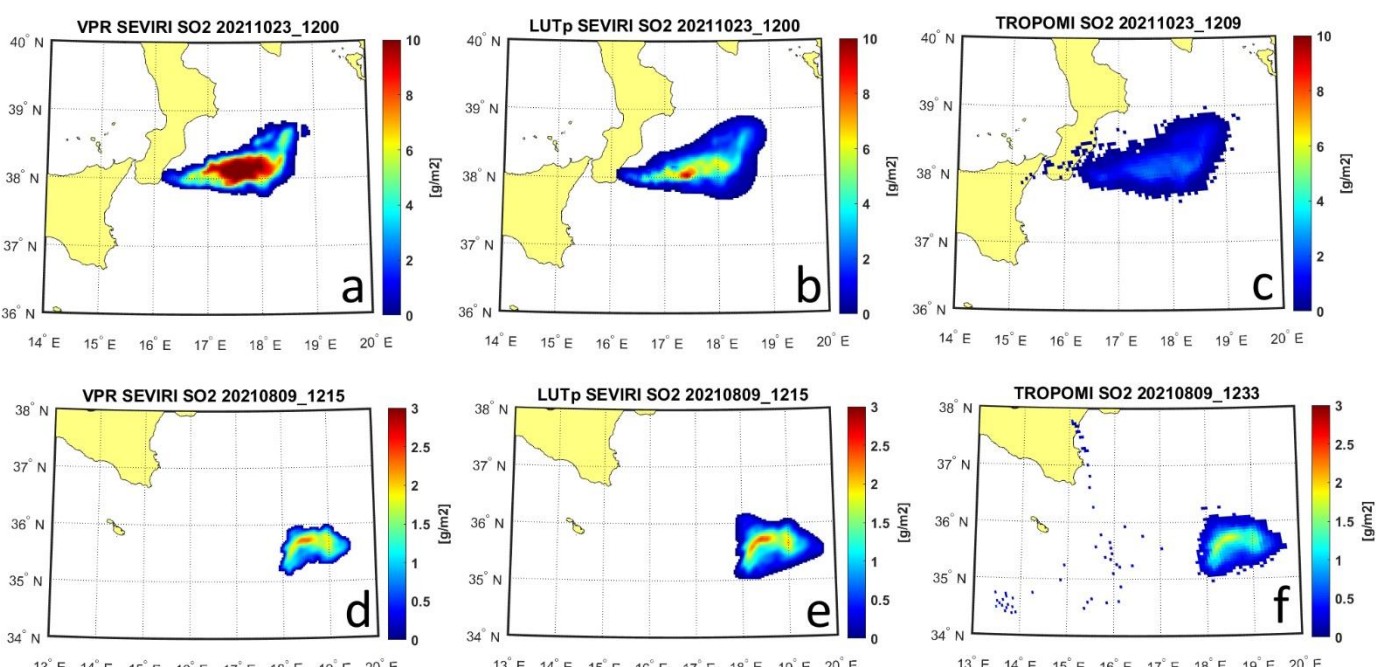

**Figure 14.** SO$_2$ column densities [g m$^{-2}$] from SEVIRI-VPR (**a**,**d**), SEVIRI-LUT$_P$, (**b**,**e**), and TROPOMI (**c**,**f**) for two LF episodes (at the top the 23 October 2021, at the bottom the 9 August 2021). Note that the color ramp for the two LFs is not the same.

## 4. Conclusions

In this work, the characterization of the 66 paroxysmal explosive episodes that occurred at Etna volcano in the period 2020–2022 has been presented. This analysis was performed by processing the satellite images collected from the SEVIRI sensor on board MSG geostationary satellite every 15 min, to retrieve the volcanic cloud top height (VCTH), the ash, ice, and SO$_2$ total masses' time series as well as the cumulative masses for the whole period.

The VCTHs were computed by exploiting three different algorithms ("Darkest Pixel", "Cloud Tracking", and "HYSPLIT"). To compute the most reliable VCTH value for each event, a weighted average procedure has been developed considering the uncertainties and limits of applicability of the different methods. The values obtained vary between 4 and 13 km, and the entire period can be approximately divided into three main time ranges characterized by average VCTH values of 9 (from 13 December 2020 to 19 March 2021), 6 (from 24 March 2021 to 17 June 2021) and 10 km (from 19 June 2021 to 21 February 2022). A very similar trend (Pearson's correlation coefficient of 0.65) is found in Calvari and Nunnari [24] which processed the measurements collected from the ground-based TIR cameras to retrieve the maximum LF heights. The results obtained were compared with the maximum volcanic cloud top altitude reported in the VONA bulletin and with SO$_2$ center of mass heights obtained from TROPOMI data. Despite the very different methods, the results show the same trend and are in good agreement.

The volcanic cloud detection and the discrimination between ash and ice in the volcanic cloud have been realized by exploiting the BTD procedure. The thresholds needed for the discrimination are computed, for each LF, by using RTM simulations performed with MODTRAN 5.3 and considering the SEVIRI spectral characteristics, the VCTH, and the specific atmospheric conditions. These values range from 0.48 to 1.83, with a mean value of 1.04, which can be used as a first approximation for Etna to avoid RTM computations.

More than two thousand SEVIRI images were processed and, for each of them, the corresponding ash/ice effective radius, aerosol optical depth, columnar abundance, and SO$_2$ columnar abundance maps were obtained. Trends of ash/ice/SO$_2$ total mass are in good correlation with VCTH and in most cases, the ice content is greater than ash. Due to

the generally very short, but intense, nature that characterized these LFs, the maximum values of the total masses can be considered a good estimate of the quantities of particles and SO$_2$ emitted into the atmosphere by the volcano, as the dilution effect is negligible.

It is also interesting to note that the logarithm of the ash/ice/SO$_2$ total masses' trends are in good agreement (Pearson's correlation coefficient of 0.57) with the lava deposits obtained by processing the same SEVIRI images [61].

A good agreement is also found between SEVIRI VPR and LUT$_p$ ash and ice retrievals, while VPR results are significantly bigger than LUT$_p$ for SO$_2$ when high ice content is present in the volcanic cloud.

Table 1 reports the cumulative and the average ash/ice/SO$_2$ total masses considering all the LF episodes as also the three main periods identified from VCTHs retrievals. It is important to remark that the characteristics of most Etna 2020–2022 LFs (short episodes but sometimes very energetic) make the quantitative estimation of the various parameters critical. The formation of volcanic clouds rich in ash and ice (opaque pixels) makes the retrieval errors greater than in other circumstances. Critical is the cross-comparison with S5P-TROPOMI due to the different spectral characteristics of the two sensors (TIR vs. UV). In this case, the correlation is good only when ash and ice have low value contents. As expected, when high ash and ice values are present, SEVIRI SO$_2$ retrievals are greater than those found with TROPOMI.

**Table 1.** Cumulative and average (in brackets) ash/ice/SO$_2$ total masses in the overall period (OP) December 2020–February 2022 (upper row) and the three specific periods identified (P1, P2, and P3), both from VPR and LUT$_p$ procedures.

| Date | Ash [Gg] | | Ice [Gg] | | SO$_2$ [Gg] | |
|---|---|---|---|---|---|---|
| | **VPR** | **LUTp** | **VPR** | **LUTp** | **VPR** | **LUTp** |
| OP: 13-12-2020–21-02-2022 | 751 (12.7) | 773 (13.1) | 2291 (38.8) | 2272 (38.5) | 666 (11.3) | 518 (8.8) |
| P1: 13-12-2020–19-03-2021 | 437 (21.8) | 413 (20.7) | 1539 (77.0) | 1413 (70.7) | 352 (17.6) | 226 (11.3) |
| P2: 24-03-2021–17-06-2021 | 18 (1.2) | 42 (2.8) | 69 (4.6) | 91 (6.1) | 35 (2.3) | 48 (3.2) |
| P3: 19-06-2021–21-02-2022 | 296 (12.3) | 317 (13.2) | 683 (28.4) | 768 (32.0) | 279 (11.6) | 244 (10.2) |

The retrieval of ice particles produced in the volcanic cloud is another valuable result of this work. For almost all the LFs, the ice production is relevant and the mass is generally higher than the mass of the ash particles. From the ice mass and knowing the amount of atmospheric water vapor at the height of the cloud, it should be possible to compute the amount of water vapor emitted during the eruption. These results could be used to try to understand, in detail, the mechanisms through which both atmospheric and volcanic water vapor, together with ash particles, contribute to the formation of ice particles in the volcanic cloud.

**Supplementary Materials:** The following supporting information can be downloaded at: https://www.mdpi.com/article/10.3390/rs15082055/s1, Figure S1: The complete ash/ice/SO$_2$ time series plots for all the 2020–2022 Etna Lava Fountains using the VPR procedure.

**Author Contributions:** Conceptualization and work coordination, L.G. and S.C.; methodology, L.G. and S.C.; SEVIRI data processing, L.G. and D.S.; TROPOMI data processing, N.T.; data analysis, L.G., S.C., and N.T.; writing—original draft preparation, L.G.; writing—review and editing, L.G., S.C., D.S., N.T., and L.M.; Funding acquisition, L.M. All authors have read and agreed to the published version of the manuscript.

**Funding:** This study has been developed in the framework of the INGV project Pianeta Dinamico Tema 8-ATTEMPT-2021 financed by Italian MUR ("Fondo finalizzato al rilancio degli investimenti delle amministrazioni centrali dello Stato e allo sviluppo del Paese, legge 145/2018"); the IMPACT project funded by the Istituto Nazionale di Geofisica e Vulcanologia ("Vulcani" Department). D59C19000130005).

**Data Availability Statement:** Most of the data presented in this study is already available in the article and supplementary material itself. The remaining data are available on request from the corresponding author.

**Acknowledgments:** The authors gratefully acknowledge the NOAA Air Resources Laboratory (ARL) for the provision of the HYSPLIT transport and dispersion model and READY website (https://www.ready.noaa.gov) used in this publication.

**Conflicts of Interest:** The authors declare no conflict of interest.

## List of Acronyms

| | |
|---|---|
| $AOD_{0.55}$ | Aerosol optical depth at 550 nm |
| BT | Brightness temperature |
| BTD | Brightness Temperature Difference |
| CT | Cloud Tracking method |
| COBRA | Covariance-Based Retrieval Algorithm |
| DP | Darkest pixel method |
| HYSPLIT | Hybrid Single-Particle Lagrangian Integrated Trajectory model |
| HY | HYSPLIT method |
| INGV | Istituto Nazionale di Geofisica e Vulcanologia |
| LF | Lava fountain |
| $LUT_p$ | Look-Up Tables procedure |
| m | Columnar abundance density |
| M | Total mass |
| MAST | Multimission Acquisition SysTem |
| MSG | Meteosat Second Generation geostationary satellite |
| NSEC | New Southeast Crater |
| PTH | Pressure-Temperature-Humidity |
| PW | Precipitable water |
| $R_e$ | Effective radius |
| RTM | Radiative Transfer Models |
| S5P | Sentinel-5 Precursor |
| SEVIRI | Spinning Enhanced Visible and InfraRed Imager |
| $SO_2$ | Sulphur dioxide |
| SST | Sea surface temperature |
| TIR | Thermal InfraRed |
| TROPOMI | TROPOspheric Monitoring Instrument |
| VCTH | Volcanic cloud top height |
| VONA | Volcano Observatory Notices for Aviation bulletin |
| VPR | Volcanic Plume Retrieval procedure |
| VZA | View Zenith Angle |

## Appendix A

**Table A1.** List of all the 2020–2022 lava fountains considered. In the first column, the LF numeration according to Calvari and Nunnari [24] is reported. Date and time (2nd and 3rd columns) refer to the time at which the maximum of VCTH from the DP method was obtained; 10 m and 30 s were added to SEVIRI starting acquisition time to account for the position of Etna in the "Earth Full-Disk" image. The 4th column indicates the mean weighted VCTH obtained by applying Equations (1)–(3). Finally, columns 5th–10th report the maximum ash/ice/$SO_2$ total mass values obtained from VPR and $LUT_p$ procedures applied to SEVIRI images. For $SO_2$ only, the maximum in the time range between the start and the end (+1 h) of the eruption was considered.

| LF n. | Date | Time [UTC] | VCTH [km] | $SO_2$ Mass [Gg] | | Ash Mass [Gg] | | Ice Mass [Gg] | |
|---|---|---|---|---|---|---|---|---|---|
| | | | | VPR | $LUT_p$ | VPR | $LUT_p$ | VPR | $LUT_p$ |
| 1–3 | 13-12-2020 | 22:25:30 | 7.0 | 1.6 | 3.0 | 0.8 | 1.8 | 2.2 | 3.2 |

**Table A1.** *Cont.*

| LF n. | Date | Time [UTC] | VCTH [km] | SO$_2$ Mass [Gg] | | Ash Mass [Gg] | | Ice Mass [Gg] | |
|---|---|---|---|---|---|---|---|---|---|
| | | | | VPR | LUT$_p$ | VPR | LUT$_p$ | VPR | LUT$_p$ |
| 4 | 21-12-2020 | 09:25:30 | 11.4 | 18.7 | 13.3 | 16.5 | 13.7 | 44.1 | 59.8 |
| 5 | 22-12-2020 | 05:10:30 | 6.5 | 5.2 | 5.3 | 2.7 | 6.9 | 6.2 | 14.6 |
| 6 | 18-01-2021 | 20:25:30 | 4.0 | 6.7 | 6.0 | 1.8 | 5.0 | 10.3 | 21.2 |
| 7 | 16-02-2021 | 16:55:30 | 10.1 | 9.0 | 6.3 | 11.7 | 10.4 | 22.3 | 21.5 |
| 8 | 18-02-2021 | 00:25:30 | 9.2 | 7.9 | 4.8 | 3.9 | 4.9 | 16.6 | 18.2 |
| 9 | 19-02-2021 | 09:40:30 | 11.0 | 18.4 | 10.1 | 19.0 | 13.4 | 141.9 | 107.8 |
| 10 | 21-02-2021 | 00:55:30 | 11.4 | 13.9 | 7.8 | 13.8 | 13.7 | 57.4 | 56.4 |
| 11–12 | 22-02-2021 | 23:55:30 | 11.6 | 24.8 | 10.6 | 26.5 | 24.6 | 119.6 | 83.2 |
| 13 | 24-02-2021 | 20:55:30 | 11.2 | 12.0 | 9.7 | 13.1 | 12.8 | 65.6 | 61.9 |
| 14 | 28-02-2021 | 08:25:30 | 11.1 | 18.5 | 16.1 | 26.5 | 20.6 | 84.2 | 62.3 |
| 15 | 02-03-2021 | 14:40:30 | 8.9 | 20.1 | 13.6 | 3.1 | 5.2 | 39.2 | 35.2 |
| 16 | 04-03-2021 | 02:55:30 | 4.7 | 0.7 | 0.9 | 0.9 | 1.2 | 0.6 | 0.7 |
| 17 | 04-03-2021 | 08:40:30 | 11.6 | 15.0 | 6.9 | 21.4 | 19.1 | 120.3 | 64.4 |
| 18 | 07-03-2021 | 06:55:30 | 11.6 | 22.8 | 15.6 | 36.0 | 28.4 | 122.8 | 88.9 |
| 19 | 10-03-2021 | 01:40:30 | 9.1 | 54.8 | 20.2 | 89.9 | 72.7 | 219.6 | 208.2 |
| 20 | 12-03-2021 | 09:25:30 | 10.8 | 21.8 | 15.0 | 22.3 | 17.4 | 108.0 | 91.8 |
| 21 | 15-03-2021 | 00:55:30 | 4.7 | 16.4 | 10.6 | 11.0 | 21.3 | 20.0 | 38.1 |
| 22 | 17-03-2021 | 03:10:30 | 5.1 | 14.6 | 15.1 | 2.8 | 5.7 | 20.2 | 28.9 |
| 23 | 19-03-2021 | 09:55:30 | 10.1 | 49.4 | 29.1 | 113.1 | 114.6 | 318.2 | 347.0 |
| 24 | 24-03-2021 | 03:40:30 | 5.6 | 5.8 | 9.3 | 2.8 | 6.6 | 7.9 | 16.5 |
| 25 | 01-04-2021 | 00:55:30 | 6.9 | 4.6 | 1.8 | 2.9 | 11.7 | 15.0 | 2.6 |
| 26 | 19-05-2021 | 00:40:30 | 5.7 | 0.6 | 5.6 | 0.3 | 1.2 | 3.1 | 6.8 |
| 27 | 21-05-2021 | 03:40:30 | 5.6 | 0.9 | 3.1 | 1.1 | 2.3 | 5.2 | 4.0 |
| 28 | 22-05-2021 | 21:40:30 | 6.4 | 0.4 | 1.5 | 1.4 | 2.6 | 0.9 | 2.7 |
| 29 | 24-05-2021 | 21:25:30 | 7.2 | 0.9 | 1.8 | 1.9 | 3.2 | 0.3 | 0.6 |
| 30 | 25-05-2021 | 18:55:30 | 4.3 | 1.1 | 1.8 | 0.4 | 0.8 | 0.0 | 0.0 |
| 31 | 26-05-2021 | 11:10:30 | 5.6 | 1.3 | 1.8 | 1.1 | 1.4 | 2.3 | 0.3 |
| 36 | 30-05-2021 | 05:10:30 | 6.0 | 2.4 | 1.7 | 0.8 | 1.8 | 5.9 | 8.5 |
| 37 | 02-06-2021 | 10:25:30 | 6.3 | 2.8 | 3.6 | 0.7 | 1.2 | 5.0 | 10.1 |
| 38 | 04-06-2021 | 17:25:30 | 7.1 | 1.7 | 3.2 | 1.8 | 3.6 | 8.6 | 6.3 |
| 39 | 12-06-2021 | 21:25:30 | 7.2 | 3.2 | 3.9 | 1.6 | 3.4 | 3.8 | 10.4 |
| 40 | 14-06-2021 | 21:55:30 | 5.3 | 2.7 | 4.5 | 0.7 | 1.2 | 3.1 | 3.8 |
| 41 | 16-06-2021 | 12:25:30 | 7.8 | 2.9 | 3.7 | 0.3 | 0.3 | 2.8 | 3.1 |
| 42 | 17-06-2021 | 23:40:30 | 6.0 | 3.6 | 2.9 | 0.5 | 1.2 | 5.3 | 15.6 |
| 43 | 19-06-2021 | 19:40:30 | 10.3 | 2.8 | 3.6 | 4.3 | 6.0 | 1.0 | 1.4 |
| 44 | 20-06-2021 | 23:40:30 | 10.2 | 4.0 | 3.4 | 5.9 | 5.6 | 9.9 | 13.0 |
| 45 | 22-06-2021 | 03:40:30 | 10.8 | 5.7 | 5.0 | 11.1 | 13.2 | 7.8 | 17.7 |
| 46 | 23-06-2021 | 03:25:30 | 10.9 | 6.4 | 6.0 | 10.9 | 11.1 | 10.2 | 12.6 |
| 47 | 23-06-2021 | 19:10:30 | 11.1 | 4.2 | 2.2 | 5.5 | 5.5 | 14.3 | 23.2 |
| 48 | 24-06-2021 | 10:40:30 | 11.4 | 7.4 | 4.1 | 5.5 | 4.9 | 29.9 | 35.7 |

**Table A1.** *Cont.*

| LF n. | Date | Time [UTC] | VCTH [km] | SO₂ Mass [Gg] | | Ash Mass [Gg] | | Ice Mass [Gg] | |
|---|---|---|---|---|---|---|---|---|---|
| | | | | VPR | LUT$_p$ | VPR | LUT$_p$ | VPR | LUT$_p$ |
| 49 | 25-06-2021 | 01:40:30 | 9.0 | 1.0 | 1.3 | 1.9 | 2.6 | 1.0 | 1.9 |
| 50 | 25-06-2021 | 19:25:30 | 9.2 | 2.2 | 2.8 | 3.2 | 4.9 | 1.4 | 3.2 |
| 51 | 26-06-2021 | 16:40:30 | 11.4 | 2.9 | 2.0 | 4.1 | 9.7 | 10.6 | 14.0 |
| 52 | 27-06-2021 | 09:40:30 | 10.1 | 4.0 | 2.5 | 5.6 | 6.0 | 26.4 | 44.9 |
| 53 | 28-06-2021 | 15:10:30 | 11.5 | 7.8 | 6.1 | 11.1 | 10.6 | 34.1 | 46.2 |
| 54 | 01-07-2021 | 23:40:30 | 5.8 | 14.5 | 17.2 | 3.8 | 5.9 | 1.6 | 0.0 |
| 55 | 04-07-2021 | 16:25:30 | 6.8 | 7.8 | 9.9 | 4.3 | 8.0 | 8.8 | 14.0 |
| 56 | 06-07-2021 | 23:10:30 | 8.4 | 5.6 | 8.0 | 2.2 | 4.1 | 1.4 | 0.1 |
| 57 | 08-07-2021 | 21:40:30 | 10.7 | 9.4 | 9.6 | 5.5 | 7.4 | 11.0 | 15.5 |
| 58 | 14-07-2021 | 12:10:30 | 10.4 | 12.4 | 12.4 | 10.7 | 14.2 | 9.8 | 10.7 |
| 59 | 20-07-2021 | 07:25:30 | 10.6 | 13.8 | 13.1 | 5.4 | 18.6 | 20.3 | 25.8 |
| 60 | 31-07-2021 | 23:10:30 | 9.6 | 12.5 | 13.4 | 9.2 | 10.0 | 18.7 | 28.0 |
| 61 | 09-08-2021 | 03:25:30 | 9.4 | 22.4 | 20.0 | 13.2 | 15.6 | 46.0 | 64.2 |
| 62 | 29-08-2021 | 16:55:30 | 8.0 | 8.5 | 10.6 | 4.0 | 6.7 | 9.2 | 14.1 |
| 63 | 21-09-2021 | 07:55:30 | 10.4 | 21.1 | 19.5 | 14.4 | 16.3 | 37.7 | 53.0 |
| 64 | 23-10-2021 | 09:25:30 | 13.0 | 52.1 | 36.0 | 86.3 | 72.0 | 176.7 | 189.4 |
| 65 | 10-02-2022 | 21:25:30 | 11.5 | 33.8 | 23.6 | 41.8 | 41.2 | 128.4 | 89.4 |
| 66 | 21-02-2022 | 12:25:30 | 11.8 | 16.5 | 10.6 | 25.8 | 17.2 | 66.5 | 49.9 |

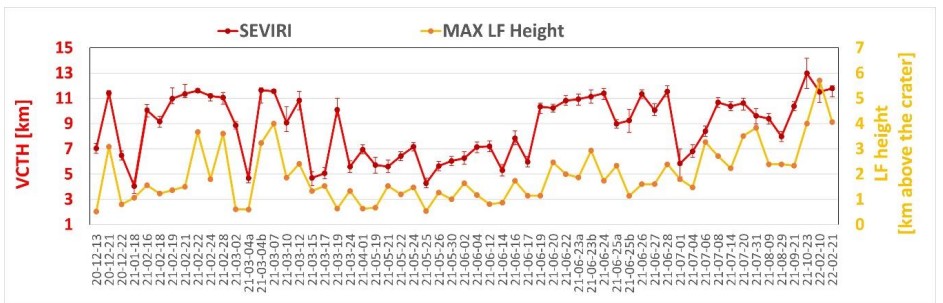

**Figure A1.** Comparison between the SEVIRI-VCTH and the maximum LF height obtained from TIR ground-based cameras by Calvari and Nunnari [24].

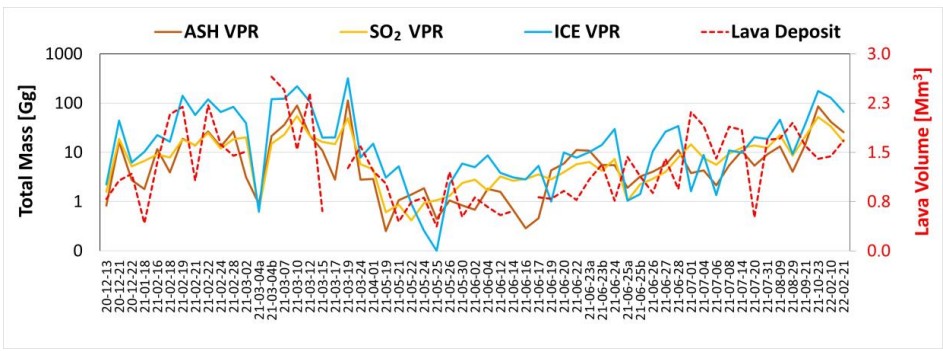

**Figure A2.** Comparison between VPR Ash/Ice/SO₂ total masses with the ground lava deposits obtained by Ganci et al. [61] from SEVIRI images.

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
