# Peer review of "Volcanic Clouds Characterization of the 2020–2022 Sequence of Mt. Etna Lava Fountains Using MSG-SEVIRI and Products’ Cross-Comparison"

_remotesensing, doi:10.3390/rs15082055_

Round 1

Reviewer 1 Report

you need to convert tonnes into the international system

section 2.1 "In this work we compare the VCTH obtained from 3 different methods, all applied to SEVIRI images: darkest pixel, cloud tracking, and HYSPLIT" However, HYSPLIT has major limitations for this type of application (afor example it does not consider massive particles), therefore the height calculated in this way is very different from that of a particle emitted by the volcano. Review this part and describe the limitations of HYSPLIT

figures 7, 10, 11, 13: x-axis is not clear, you need to better organize the figure including the year. Also, how are error bars calculated?

equations 1-3: how were they derived?

Minor revisions:

often the year is not indicated (example page 13 "the period from 16 February to 19 March was characterized by high volcanic plumes"). Review

sometimes months are shortened sometimes not, (Feb - February ?). Review

Reviewer 2 Report

Summary

The manuscript “Volcanic clouds characterization of the 2020-2022 sequence of Mt. Etna lava fountains using MSG-SEVIRI and products cross-comparison” presents the analysis of a series of >60 lava fountain events that occurred at Mount Etna in 2020-2022, with thermal infrared images acquired by the SEVIRI instrument. The manuscript presents masses of SO2, ash and ice retrieved from the satellite data for each event, and compares the MSG retrievals with other methods. The analysis is carried out with rigour, presented in a clear manner, and the results should be of very high interest to the readers of Remote Sensing. Although the quantitative characterization of events is sufficient for publication, I would like to suggest that the manuscript could benefit from a more extensive discussion of the results. 

I have outlined my comments below. The lack of line numbers makes it difficult to refer to specific statements but I attempted to quote each statement at the start of each comment. 

General comments

How were the weighted average coefficients chosen for equations (1), (2) and (3)? Further to this point, the manuscript reports “the 3 methods agree very well and the mean absolute difference with the weighted mean (VCTH) is 0.6, 1.0, and 0.6 km for DP, CT, and HY respectively.”. What is the significance of this mean absolute difference value? Is it not a representation of the weighting coefficients used rather than a measure of how well the values agree with each other? This is a pretty important point, given that the VCTH time series and its agreement with VONA and TROPOMI is a central finding of this manuscript. It merits further discussion. 

3 parameters are retrieved: Re [μm], AOD0.55, and ma/mi/mSO2 [g/m2]. The manuscript describes the LF events in terms of the total mass, but never explores any potential changes in particle sizes, or changes in the maximum/mean/median columns and their distribution. This may be a missed opportunity to exploit what is a fantastic and rich dataset. For example, were there any variations in the retrieved Re for the 3 periods described here? Is the spatial extent of a given LF plume or the maximum column related the duration of the event, or the VCTH? Exploring the relationship between the individual quantities retrieved from the dataset would be a great addition to this work, and provide material for further discussion of the results. 

One particular outstanding question to me is in regards to plume segmentation. Using the BTD method, individual pixels are identified as either ash OR ice. This leads to an artificial “segmentation” of the plume into an ash portion and an ice portion, which is well illustrated in the image presented in Figure 9. Ash is identified in the centre of the plume, and ice at the edges. Because both species are likely to be present in the plume at the same time, this represents a limitation in the method. The ice content in the concentrated ashy portion of the plume is essentially ignored. And if ash particles act as nuclei for ice, it stands to reason that there may be a large quantity of ice not detected with this method. Could the manuscript explore this question further and discuss how this effect impacts the estimates of both ash and ice masses?

The discrepancies observed here between the IR and UV retrievals for SO2 are particularly interesting and should be explored further. The SEVIRI dataset offers the possibility to explore the temporal evolution of the plume in much greater detail than the TROPOMI data. Is the total mass of SO2 retrieved with SEVIRI (with either method) ever comparable to the one retrieved with TROPOMI? Perhaps the values converge when looking at downwind plumes where the ash is more dilute and therefore the effects on both methods are less pronounced? It would be helpful to add a figure with concurrent (or close in time) images of TROPOMI and SEVIRI SO2 retrievals for one or two LF events, to see how they compare in terms of spatial extent of the plume, distribution of the column amounts and whether the pixels with the highest discrepancies are indeed those with high ash/ice content. 

Specific comments

Figure 2b: VCTH over the event goes from ~3-4 km at the beginning (I assume before the onset of the LF) to ~0-1 km after the event. Given the altitude of the crater (~2.5-3 km if I remember correctly?), how would you interpret the low values of VCTH after the event? Is it possible that this apparent negative slope underlying the event reflects the fact that the atmospheric profile used to evaluate VCTH is the same over the entire time series, and therefore only representative of the thermal profile of the atmosphere for a limited number of images? If so, would a simple linear interpolation between atmospheric profiles (essentially creating a unique profile for each time step) yield more consistent results? I understand the evolution of the thermal atmospheric profile is unlikely to be linear between 6h intervals. However, it may improve VCTH accuracy in cases when the maximum height occurs several hours before or after the vertical profile was acquired. 

Section 2.1.3: How is the “best” trajectory evaluated? Are you also using the centre of gravity of the plume in the SEVIRI image, and then calculate the shortest distance to the end point of each trajectory? In Figure 4, it looks like the affected plume area encompasses 4 adjacent grid points in what I assume is the 2.5x2.5 grid of the atmospheric model. Do you consider it a match if the Hysplit trajectory endpoint is any of those 4 grid points? Are there any instances with more than one valid trajectory (for example when wind speed shows low or no wind shear over several heights)? If so, how do you assign the final VCTH?

Section 2.2: The manuscript states that “the volcanic cloud detection was performed by visual inspection”. How exactly is this done? For example, are individual pixels flagged as plume for further processing? Or is a polygon area (ROI) drawn around the plume (if so, is it a rectangle, or a more complex shape)? 

Figure 6b: The influence of VZA is unclear on this figure. I understand that it is represented by the error bars. However, from the figure as presented, it is impossible to determine of higher VZA result in higher BTD threshold, or if the opposite effect is true. Similarly, it is not possible to tell if the effect of VZA is linear or not. 

“Because of the ash/ice particles’ absorption in the whole TIR spectral range, their quantitative estimations are taken into account to correct for the SO2 amounts”. How is this done exactly? Is the OD corresponding to the retrieved amount of ash/ice calculated at 8.6 um, and then subtracted from the measured OD at 8.6 um? 

Figure 7: The three periods reported in the manuscript are not immediately obvious in this figure. It may be helpful to I them explicitly (for example with shaded areas). As well, the x-axis is somewhat confusing. The use of mm-dd format over a period covering multiple years can make it difficult to see where in the sequence each event is occurring. And the considerations made later in the manuscript about each phase of activity are not immediately apparent when presenting the LFs spaced evenly. Might I suggest using a linear time axis (like the one in Figure 8) to give the reader a better sense of how frequent the events were in each phase, and whether there are pauses between each phase? 

The average difference between VCTH and LF maximum heights is 3.5 km (considering that NSEC is about 3.3 km asl high). Given that the values reported in Calvari and Nunnari are heights above the crater, it should be straightforward to correct this dataset with the known altitude of the crater and compare VCTH with actual fountain heights above sea level in Figure A2. The resulting Pearson coefficient and standard deviation would be a lot more informative and could generate further discussion about the possible discrepancies between the two dataset. For example, Calvari and Nunnari base their detection on the presence of hot pixels if I remember correctly. In such a case, I would expect their heights to be lower than the VCTH estimated here, which assumes that the plume reaches thermal equilibrium with the surrounding atmosphere. 

Regarding the VZA, a fixed value was considered for a geostationary SEVIRI sensor (VZA = 45°, typical of the Etna area). In this way for each LF considered, only one BTD threshold value was used. Figure 6b shows pretty significant variations of the BTD with VZA, and the manuscript mentions that typical values for Etna area range from 35 to 55 degrees. Moreover, the plume shown in Figure 4b for example, has travelled quite far from Etna. Is a VZA of 45 degree still relevant for this plume? In general, could you add a discussion on the expected effect of changing VZA on the BTD threshold, and in which direction you would expect this effect to go (under- or overestimation)? On a more general note, would it be feasible to assign a BTD threshold to each pixel (or to an area of pixels defined by the 2.5x2.5 degree grid of the atmospheric model)? Or would this increase the computing time to a point where it is prohibitive?

Figure 10: Here as well, it is difficult to observe the influence of seasonality on the retrieved ice mass. A linear time scale would be more appropriate (as in Figure 8). 

Despite the significant difference between the two quantities, the two trends show quite good agreement. What is meant by this statement? What are the significant differences? And how can the agreement be good if there are significant differences? Please elaborate. 

Figure 12. It would be helpful to see scatter plots of TROPOMI vs VPR and TROPOMI vs LUTp. 

Figure 13. Are the total masses plotted for SEVIRI the maximum total masses for the event? Or the total mass from an individual SEVIRI image at the time closest to the TROPOMI overpass? It would be helpful to present an example of TROPOMI vs SEVIRI images (similar to Figure 9) with all 3 methods used for SO2. Is the spatial distribution of the cloud comparable? 

Minor language typos and suggestions

For example, it should be noted that despite the VCTH of 23 October 2021 is greater than that of 21 September 2021, the BTD threshold behaves oppositely. Please consider rewording this sentence. Suggest replacing with “despite the fact that the VCTH…”, and explain explicitly what is meant by “the threshold behaves oppositely”, for example, “…the BTD threshold is higher, despite the fact that the VCTH is higher”. 

“as documented by several works [54]”: Rather than citing several individual studies, this reference is a multi-disciplinary effort, which contains a review of previous findings. 

Reviewer 3 Report

Manuscript focuses on Etna eruption monitoring using SEVIRI data and cross validation with other ancillary data for the period 2020-2022. Manuscript is interesting however should be better organized. Especially parts of section 4 should be included in section 2. Please pay an attention to order the research clearly.

Detailed comments:

Fig. 1a. Please provide a/satellite /aerial/DEM (?) images’ sources.

Fig. 1b. It is not seen a lava fountain in the photo? Please enlarge the photo or provide more illustrative one.

Section. 2. How many SEVIRI images did authors process in total? What were an initial and a final dates of monitoring?

Fig.2. Is DP procedure or method, please clarify. Please note that procedures are routine steps and rigid. Methods are more flexible and have a narrower scope. This comment applies to the entire manuscript. Authors use the words “procedure” and “method” as synonyms but they are not.

Fig.2. Please provide clarification on the legend symbols (the same for figure 3). Please find the order key presenting the figures (If 2a and 2b then 3b and 3a).

Line 204 – It is out of the scope of the subsection. Please compare/discuss the methods in other section.

Fig.4 The figure is not informative, please correct. Figure 4a should be enlarged.

Line 228 – Please provide a date of RGB image.

Line 253 – Does BTD approach relate to 28.02.2021 or 18.02.2021 image? Please clarify and order the manuscript.

Figure 6b– This is unclear how PTH and SST are illustrated. Please clarify.

Lines 303–304– Please improve the English style.

Lines 317–322–These are conclusions. Please reorganize the manuscript.

Section 4 – Cross-comparison using other ancillary data is also a method. Authors did not include it in the method section. Please reorganize and order manuscript.

Table 1 should be moved to results section.
